# Lack of Cathepsin D in the Renal Proximal Tubular Cells Resulted in Increased Sensitivity against Renal Ischemia/Reperfusion Injury

**DOI:** 10.3390/ijms20071711

**Published:** 2019-04-05

**Authors:** Chigure Suzuki, Isei Tanida, Masaki Ohmuraya, Juan Alejandro Oliva Trejo, Soichiro Kakuta, Takehiko Sunabori, Yasuo Uchiyama

**Affiliations:** 1Department of Cellular and Molecular Neuropathology, Juntendo University Graduate School of Medicine, Bunkyo-Ku, Tokyo 113-0033, Japan; cgsuzuki@juntendo.ac.jp (C.S.); oliva@juntendo.ac.jp (J.A.O.T.); skakuta@juntendo.ac.jp (S.K.); 2Department of Cellular and Molecular Pharmacology, Juntendo University Graduate School of Medicine, Bunkyo-Ku, Tokyo 113-0033, Japan; 3Department of Cell Biology and Neuroscience, Juntendo University School of Medicine, Bunkyo-Ku, Tokyo 113-0033, Japan; tsunaho@juntendo.ac.jp; 4Department of Genetics, Hyogo College of Medicine, Nishinomiya 663-8131, Japan; ohmuraya@hyo-med.ac.jp; 5Laboratory of Morphology and Image Analysis, Biomedical Research Center, Juntendo University Graduate School of Medicine, Bunkyo-Ku 113-0033, Japan

**Keywords:** cathepsin D, renal proximal tubular cells, ischemia/reperfusion injury, gene knockout mouse

## Abstract

Cathepsin D is one of the major lysosomal aspartic proteases that is essential for the normal functioning of the autophagy-lysosomal system. In the kidney, cathepsin D is enriched in renal proximal tubular epithelial cells, and its levels increase during acute kidney injury. To investigate how cathepsin D-deficiency impacts renal proximal tubular cells, we employed a conditional knockout *CtsD^flox/−^; Spink3^Cre^* mouse. Immunohistochemical analyses using anti-cathepsin D antibody revealed that cathepsin D was significantly decreased in tubular epithelial cells of the cortico-medullary region, mainly in renal proximal tubular cells of this mouse. Cathepsin D-deficient renal proximal tubular cells showed an increase of microtubule-associated protein light chain 3 (LC3; a marker for autophagosome/autolysosome)-signals and an accumulation of abnormal autophagic structures. Renal ischemia/reperfusion injury resulted in an increase of early kidney injury marker, Kidney injury molecule 1 (Kim-1), in the cathepsin D-deficient renal tubular epithelial cells of the *CtsD^flox/−^; Spink3^Cre^* mouse. Inflammation marker was also increased in the cortico-medullary region of the *CtsD^flox/−^; Spink3^Cre^* mouse. Our results indicated that lack of cathepsin D in the renal tubular epithelial cells led to an increase of sensitivity against ischemia/reperfusion injury.

## 1. Introduction

Cathepsin D is a major lysosomal aspartic protease that is responsible for the degradation of proteins and organelles via the autophagy-lysosomal system [1,2]. *CtsD*, which encodes cathepsin D, is indispensable for life after birth. *CtsD* gene knockout mice die at approximately postnatal day 26, exhibiting a small amount of intestinal necrosis, and a neuropathologic defect that resembles the phenotype of neuronal ceroid-lipofuscinosis (known as *CLN10*) [3,4,5,6,7]. Cathepsin D-deficient neurons exhibit autofluorescence, and accumulate subunit c of mitochondrial ATPase in the lysosomes as a lipofuscin [8]. In addition, cathepsin D-deficient neurons accumulate abnormal autophagy-related structures and an increase of LC3 (a marker for autophagosome/autolysosome)-signals [4,5,8].

In the kidney, cathepsin D is significantly expressed in proximal tubular cells. Renal ischemia/reperfusion is a common cause of acute kidney injury [9]. Particularly, proximal tubular epithelial cells of the cortico-medullary region are highly susceptible to acute ischemia [10,11,12,13]. Injury causes a rapid loss of cytoskeletal integrity and epithelial cell polarity with shedding of the brush border, apoptosis, and necrosis [14]. Using a mouse model of renal ischemia/reperfusion injury, it has been shown that cathepsin D is up-regulated in damaged renal proximal tubular epithelial cells [15,16]. However, little is known about the contribution of cathepsin D in the renal proximal tubular epithelial cells. To clarify the function of cathepsin D in the renal proximal tubular epithelial cells, we investigated cathepsin D-deficient renal proximal tubular epithelial cells using a *CtsD^flox/−^; Spink3^Cre^* mouse model. 

## 2. Results

### 2.1. Cathepsin D-Deficiency Caused an Increase of LC3-Positive Signals in Renal Proximal Tubular Epithelial Cells

To investigate the function of cathepsin D in the renal proximal tubular cells, we employed a *CtsD^flox/−^; Spink3^Cre^* mouse model. *CtsD^flox/−^* mice are heterozygous for a loxP-flanked exon2 of *CtsD* gene and lack of exon2 of *CtsD* gene on chromosome 7. A loxP-flanked allele is deleted by a *Cre* recombinase. *CtsD ^flox/−^* mice were crossed with *Spink3^Cre^* mice to produce *CtsD^flox/−^; Spink3^Cre^* mice. In *Spink3^Cre^* mouse, a *Cre* recombinase expresses in acinar cells of the pancreas, kidney, lung, and a small proportion of cells in the gastrointestinal tract and liver [17]. Accordingly, in *CtsD^flox/−^; Spink3^Cre^* mice, the exon2 of *CtsD* gene is deleted in those tissues. 

During an observation period of 12 months, *CtsD^flox/−^; Spink3^Cre^* mice grew normally and were fertile, while *CtsD* gene knockout mice die at approximately postnatal day 26. We first investigated whether or not cathepsin D is depleted in the proximal tubular cells of the cortico-medullary region in the *CtsD^flox/−^; Spink3^Cre^* mouse kidney (Figure 1). We used anti-cathepsin D and anti-Megalin (a marker for proximal tubular cells) antibodies, to investigate endogenous cathepsin D in the renal tissues from *CtsD^flox/flox^* and the *CtsD^flox/−^; Spink3^Cre^*. In the *CtsD^flox/flox^* mouse tissue, cathepsin D-positive signals were recognized well in the renal tubular epithelial cells of the cortico-medullary region, mainly in the proximal tubular cells. In contrast, few signals of cathepsin D were recognized in the renal tubular epithelial cells of the cortico-medullary region of the *CtsD^flox/−^; Spink3^Cre^* mouse. 

### 2.2. Cathepsin D-Deficiency Causes an Accumulation of Abnormal Autophagic Structures in Renal Proximal Tubular Epithelial Cells

It has been reported that LC3 (a marker for autophagosome/autolysosome)-positive signals are increased in the cathepsin D-deficient neurons because of a defect in the autophagy-lysosome system [4,5,8]. We therefore investigated whether or not LC3 increased in the cathepsin D-deficient renal tubular epithelial cells (Figure 2). LC3-positive dots were significantly increased in the cathepsin D-deficient renal tubular epithelial cells, while only little expression was detected in the *CtsD^flox/flox^* cells. 

In neurons, lack of cathepsin D results in the accumulation of autophagosomes, autolysosomes, and abnormal membranous structures (autolysosome-like electron-dense structures surrounded by lamellarly and concentrically arranged endoplasmic reticula (LER)) [4,18,19]. This accumulation of autophagosomes and autolysosomes has been correlated with increases in LER due to the impairment of lysosomal function [4,19]. We therefore used electron microscopy to investigate whether or not abnormal intracellular membranous structures were accumulated in the cathepsin D-deficient renal tubular epithelial cells of the cortico-medullary region. Electron microscopic analyses of the renal tubular epithelial cells in the cortico-medullary region of the *CtsD^flox/−^; Spink3^Cre^* mouse indicated that these cells contained autolysosome-like, electron-dense structures surrounded by LER (Figure 3A). Quantification, using the point-counting method [20], revealed that LER are significantly increased in cathepsin D-deficient renal proximal tubular epithelial cells (Figure 3B). Interestingly, large vacuolar structures were observed in some cathepsin D-deficient renal proximal tubular epithelial cells (Figure 2C and Figure 3Ac).

### 2.3. Cathepsin D-Deficiency Increased the Sensitivity against Renal Ischemia/Reperfusion Injury in Renal Proximal Tubular Epithelial Cells of the CtsD^flox/−^; Spink3^Cre^ Mouse

The intracellular abnormalities were observed in the cathepsin D-deficient renal tubular cells. However, *CtsD^flox/−^; Spink3^Cre^* mice grew normally and were fertile. Urine and blood analyses showed little difference between *CtsD^flox/−^; Spink3^Cre^* mice and *CtsD^flox/flox^* mice (Supplementary Appendix A). Out of the major cathepsins (cathepsin D, B, and L), only cathepsin D is significantly increased after renal ischemia/reperfusion injury [15,16]. Considering this, it is possible that cathepsin D-deficiency affects the sensitivity of the renal proximal tubular epithelial cells against renal ischemia/reperfusion injury. To investigate this hypothesis, we employed a clinical biomarker for the diagnosis of early acute kidney injury, Kim-1 (Kidney injury molecule-1), to estimate damage of the renal proximal tubular epithelial cells. In humans, urinary Kim-1 concentration is markedly increased within 12 h following kidney injury [21]. Therefore, an increase in urinary Kim-1 serves as a clinical biomarker for the diagnosis of early acute kidney injury. Under the normal physiological conditions Kim-1 was not detected by immunostaining in kidneys of both *CtsD^flox/−^; Spink3^Cre^* and *CtsD^flox/flox^* mice (Figure 4). Twenty-four hours after ischemia/reperfusion injury, Kim-1-immunopositive signals increased significantly in the kidney of *CtsD^flox/−^; Spink3^Cre^* mice, while these were faint in the kidney from *CtsD^flox/flox^* mice (Figure 4A,C). Immunohistochemical analyses using anti-cathepsin D antibody revealed that Kim-1 increased in the cathepsin D-deficient renal proximal tubular epithelial cells (Figure 4B). It indicates that cathepsin D is deficient in the same tubular cells as the Kim-1 expressed cells. Quantification of the Kim-1 positive area in the kidney indicated that the expression level of Kim-1 increased significantly in the kidney of *CtsD^flox/−^; Spink3^Cre^* mice after the ischemia/reperfusion injury, compared with kidneys from *CtsD^flox/flox^* mice (Figure 4C). 

If an increase of Kim-1 signals in the cathepsin D-deficient renal tubular cells is due to an increased sensitivity against the ischemia/reperfusion injury, inflammatory cell-infiltration will be elicited in the cortico-medullary region of *CtsD^flox/−^; Spink3^Cre^* mouse kidneys. To investigate the infiltration of macrophage into this region, we employed a macrophage marker, F4/80. F4/80-immunopositive signals increased significantly in the cortico-medullary region of *CtsD^flox/−^; Spink3^Cre^* mouse kidneys after ischemia/reperfusion injury (Figure 4D). 

## 3. Discussion

In this study, we show evidence of the importance of cathepsin D in the renal proximal tubular epithelial cells, by characterizing a cytoprotective function following ischemia/reperfusion injury. Cathepsin D is significantly decreased in the cortico-medullary region of *CtsD^flox/−^; Spink3^Cre^* mice. Lack of cathepsin D resulted in the accumulation of abnormal autophagy-related structures in renal tubular epithelial cells. LC3-positive signals and lamellarly arranged endoplasmic reticula were accumulated in the cathepsin D-deficient renal proximal tubular epithelial cells. Under the conditions for ischemia/reperfusion injury, Kim-1 levels were significantly increased in cathepsin D-deficient renal proximal tubular epithelial cells. In addition, by immunostaining with F4/80, we detected a significant increase in macrophage activity in the cortico-medullary region of the *CtsD^flox/−^; Spink3^Cre^* mice after ischemia/reperfusion injury. Therefore, we conclude that cathepsin D plays a cytoprotective role against ischemia/reperfusion injury in the renal proximal tubular epithelial cells. 

*CtsD^flox/−^; Spink3^Cre^* mice grew normally and were fertile, while cathepsin D was depleted in the renal tubular cells of the cortico-medullary region in the mice. Urine and blood analyses showed little difference between *CtsD^flox/−^; Spink3^Cre^* mice and *CtsD^flox/flox^* mice. These results suggested that other nephron cells may compensate to maintain normal kidney function in *CtsD^flox/−^; Spink3^Cre^* mice. 

LC3-positive signals, LER, and large vacuoles were accumulated in the cathepsin D-deficient renal tubular cells. As lysosomal proteases, cathepsin D, cathepsin B and cathepsin L are abundantly expressed in the kidneys [22,23,24]. Only cathepsin D is significantly increased after renal ischemia/reperfusion injury [14]. We showed that cathepsin D-deficiency in the renal tubular cells in the cortico-medullary region results in an increase of sensitivity against renal ischemia/reperfusion injury. Therefore, an increase of cathepsin D after renal ischemia/reperfusion will contribute to protect cells against the injury.

Lack of cathepsin D in renal tubular cells leads to the accumulation of LC3-positive signals and abnormal autophagic structures, resulting in an increase of sensitivity against renal ischemia/reperfusion injury. Considering that cathepsin D plays an indispensable role in the autophagy-lysosome system in many tissues, it is possible that in the renal tubular cells, the autophagy-lysosome system itself significantly contributes to the tolerance against renal ischemia/reperfusion injury. If so, autophagy-deficiency in the renal tubular cells may show more severe phenotype by renal ischemia/reperfusion injury. To clarify this point, in future studies, we will generate and investigate a renal tubular cell-specific autophagy-deficient mice. 

## 4. Materials and Methods

### 4.1. Animal Model

The serine proteinase inhibitor Kazal type 3 (*Spink3*) gene expression is restricted to acinar cells of the pancreas, and to epithelial cells of the urinary and alimentary tracts during embryonic development. In post-partum kidneys, *Spink3* mRNA was localized mainly within the tubular epithelial cells of cortico-medullary region [25]. Therefore, *CtsD^flox/−^; Spink3^Cre^* mice experienced a deletion of the loxP-flanked *CtsD* gene in the tubular epithelial cells of the kidney (Supplementary Appendix A). *CtsD^flox/−^; Spink3^Cre^* mice were produced by M. Ohmuraya’s laboratory originally using *CtsD^flox/flox^* mice and *Spink3^Cre^* mice [17] that express *Cre* recombinase under the control of the *Spink3* gene [25]. Briefly, first, *CtsD^flox/−^* mice were produced from *CtsD* flox and B6;CBA-Tg(CAG-Cre)47Imeg [26], and then, *CtsD^flox/−^* mice were crossed with *Spink3^Cre^* mice to generate *CtsD^flox/−^*; *Spink3^Cre^* mice. These mice are normally fertile and are crossed with *CtsD^flox/flox^* mice to produce the *CtsD^flox/−^*; *Spink3^Cre^* and *CtsD^flox/flox^* littermates that served as the control. *Spink3^Cre^* mice were born at the expected Mendelian ratio, grew normally, were fertile, and no phenotypic differences were detectable between *CtsD*^flox/flox^*;Spink3**^Cre/+^* and wild-type littermates [17,27].

All animal experiments were performed in accordance with the guidelines of the Laboratory Animal Experimentation of Juntendo University (project license no.290197) and were approved by the Institutional Animal Care and Use Committee of Juntendo University (06/05/2010). 

All mice were anesthetized with an isoflurane during the experimental procedures. Renal ischemia/reperfusion injury was performed by clamping the left renal artery and vein for 20 min, for ischemia/reperfusion injury, with a small vascular clip, and then initiating reperfusion by removal of the clamp [11,28,29,30]. The left kidneys were harvested at the indicated times after reperfusion. Each of the contralateral kidneys were used as controls (I/R(−)).

### 4.2. Antibodies

Rabbit anti-LC3, and anti-cathepsin D antibodies were used as previously described [31,32]. Rat antibody against F4/80 (Bio-Rad/ AbD Serotec, Oxford, UK), mouse anti-Megalin/gp 330 (NOVUS, CO, USA), rat anti-cathepsin D and goat anti-Tim-1/Kim-1 (R&D systems Inc., Minneapolis, USA) were purchased commercially.

### 4.3. Fixation and Embedding for Light and Electron Microscopy

As previously described, after being anesthetized, mice were fixed by cardiac perfusion with 4% paraformaldehyde (PFA) for light microscopy. For electron microscopy mice were fixed with 2% paraformaldehyde and 2% glutaraldehyde buffered with 0.1 M phosphate buffer (PB), pH 7.2 [4,18]. After perfusion, renal tissues were excised from the mice and cut transversely into two pieces that were further divided longitudinally into 2 pieces for light microscopy and cut into smaller pieces for electron microscopy, then immersed into the same fixatives for a further 24 h. For light microscopy, after these tissues were embedded in paraffin and cut into 5 µm sections that were then deparaffinized and stained with Hematoxylin Eosin (HE). For electron microscopy, fixed small blocks were dehydrated with a graded series of alcohols and embedded in epoxy resin (TAAB Epon 812, EM Japan, Tokyo, Japan). Silver sections were cut with an ultramicrotome (Ultracut UCT, Leica, Nussloch, Germany) and observed with an electron microscope (JEM-1230, JOEL, Tokyo, Japan) after being stained with uranyl acetate and lead citrate. For quantification of cytoplasmic organelles such as lamellarly arranged endoplasmic reticula, fingerprints, and lysosomes in renal proximal tubular epithelial cells located in the transition between the renal cortex and the medulla (RPTECs-TCM) that correspond to segment 3 of the renal proximal tubular epithelial cells, 30 cortical fields covering the RPTECs-TCM were photographed randomly at a magnification of 8000×, and digital images were analyzed using a point-counting method established by Uchiyama and Watanabe [20]. Final data were expressed as the percent volumes of each organelle in the cytoplasm of RPTECs-TCM.

### 4.4. Immunohistochemistry

Immunohistochemistry was performed in 4% Paraformaldehyde fixed 5 μm paraffin sections. Double staining was performed with mouse anti-megalin/gp330 antibody (NOVUS) and rabbit anti-cathepsin D antibody or rabbit anti-LC3 antibody and rat-anti-cathepsin D antibody (R&D). 8 μm cryosections were stained with goat anti-Tim-1/Kim1 antibody and rat anti-F4/80 antibody (Serotec). For megalin staining, deparaffinized sections were subjected to antigen retrieval by boiling in citrate buffer (19 mM, pH 6.0) for 5 min before incubation with primary antibody. After washing, secondary antibodies were added, anti-rat Cy3, anti-rabbit-Cy3, anti-rat-Alexa488, anti-rabbit-Alexa488, and anti-mouse-Alexa488 (Molecular Probes, Invitrogen, Carlsbad, CA, USA) with 4′,6-diamidino-2-phenylindole (DAPI). Immunofluorescence of the specimens was observed using a confocal fluorescence microscope FV1000 (OLYMPUS, Tokyo, Japan).

For quantification of LC3 dots, five fields in each section from 3 independent mouse kidneys were randomly selected, and LC3 positive dots in CtsD (+) proximal tubular cells in *CtsD^flox/flox^* mouse and CtsD (−) proximal tubular cells in *CtsD^flox/−^; Spink3^Cre^* were counted. Only dots of average size were included in the quantification. The ratio of dots in proximal renal tubular cells of in *CtsD^flox/−^; Spink3^Cre^* and in *CtsD^flox/flox^* was expressed as relative LC3 dot numbers.

To determine the number of damaged regions after ischemia/reperfusion injury, morphometric analyses were performed. As for the estimation of Kim-1-positive areas, the point-counting method using a double lattice system with 1.5 cm spacing was applied to fluorescent images of Kim-1-immunopositive areas in the cryosections. For quantification, 10–12 fields in 3 to 5 longitudinal sections obtained from each experimental group were randomly selected and photographed at low magnification (×50). Final data were expressed as the percent ratios of each regional volume to the total volume estimated. The data were expressed as percent ratios of the positive area to the entire parenchymal area. A detailed method for immunostaining is provided in the supplementary information.

### 4.5. Statistical Analysis

Final data from each experiment are expressed as the mean ± SEM. Statistical comparisons between the groups were performed using a student’s *t*-test, and statistical significance was set at *p* < 0.05.

## Figures and Tables

**Figure 1 ijms-20-01711-f001:**
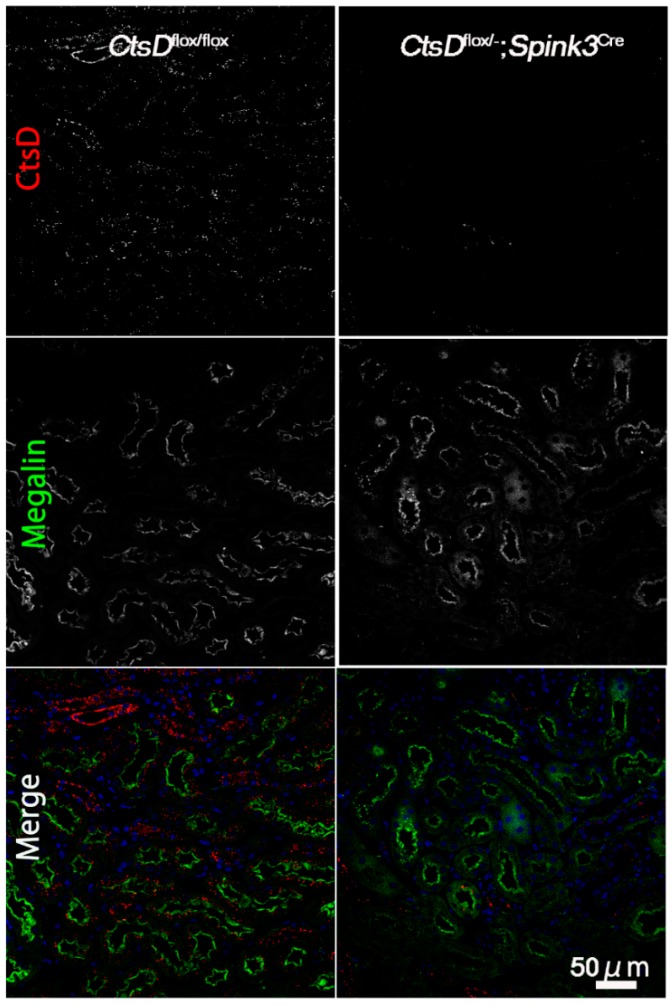
Cathepsin D is depleted in renal tubular cells in the cortico-medullary region of *CtsD^flox/−^; Spink3^Cre^* mouse. Representative images of cathepsin D (*CtsD*) staining in the cortico-medullary region of *CtsD^flox/flox^* and *CtsD^flox/−^; Spink3^Cre^* mouse kidneys. Megalin was employed as a marker of proximal tubules. In the merged images (Merge), CtsD was pseudo-colored red, Megalin was green, and 4′,6-diamidino-2-phenylindole (DAPI) was blue. Please note that in the *CtsD^flox/−^; Spink3^Cre^* mouse, cathepsin D was significantly depleted in most of the renal proximal tubular epithelial cells. Bar, 50 μm.

**Figure 2 ijms-20-01711-f002:**
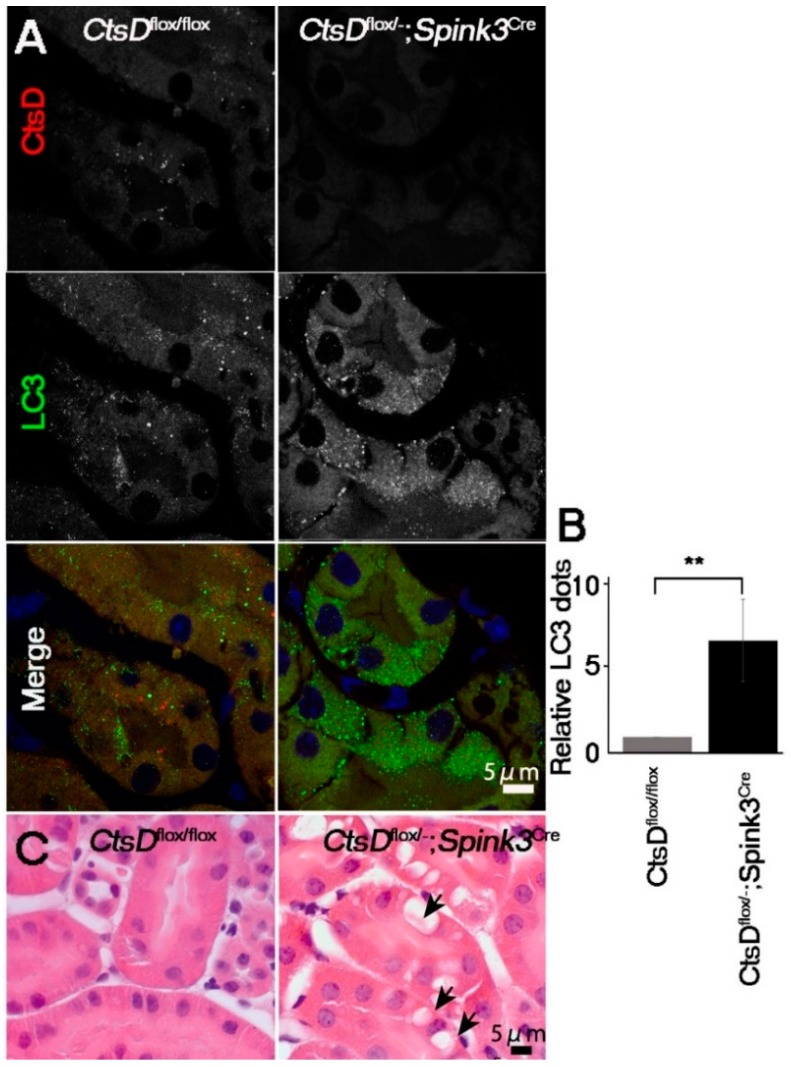
*CtsD*-deficiency resulted in an increase of LC3-positive signals and vacuoles in renal proximal tubular epithelial cells. (**A**) Representative confocal microscopic images of cathepsin D (CtsD) and LC3 immunostaining in *CtsD^flox/flox^* and *CtsD^flox/−^; Spink3^Cre^* mouse kidneys. In the *CtsD^flox/−^; Spink3^Cre^* mice, LC3 signals increased and formed granules in the cathepsin D depleted cells. Bar, 5 μm. (**B**) The number of LC3 positive dots are counted in the CtsD positive proximal tubular cells in *CtsD^flox/flox^* and CtsD negative proximal tubular cells in *CtsD^flox/−^; Spink3^Cre^* mouse. n = 3 in each group. Values are given as the mean ± SEM. ** *p* < 0.03 (**C**) Hematoxylin and eosin staining of the cortico-medullary region in *CtsD^flox/flox^* and *CtsD^flox/−^; Spink3^Cre^* mouse kidneys. Black arrows in *CtsD^flox/−^; Spink3^Cre^* indicate large vacuolar structures. Bar, 5 μm.

**Figure 3 ijms-20-01711-f003:**
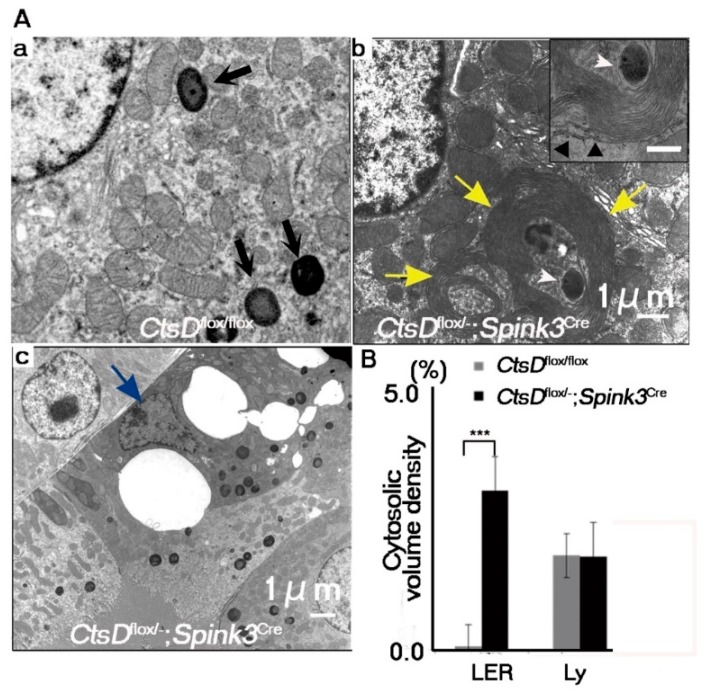
Abnormal autophagy-related structures increased in *CtsD*-deficient renal proximal tubular epithelial cells. (**A**) Electron microscopic images of *CtsD^flox/flox^* (**a**) and *CtsD^flox/−^; Spink3^Cre^* (**b**,**c**) mice. Black arrows in *CtsD^flox/flox^* (**a**) indicate normal lysosomes. Cells derived from *CtsD^flox/−^; Spink3^Cre^* contained electron-dense structures with fingerprint profiles (insert) in the apical regions (**b**). Cells derived from *CtsD^flox/−^; Spink3^Cre^* mice often possessed autophagic vacuoles and lamellarly/concentrically arranged endoplasmic reticula (LER) (**b**, yellow arrows). In some case, at the center of LER, autophagic vacuoles/autolysosomes were detected (**b**, white arrowheads). Outermost layers of LER were attached by ribosomes (**b**, insert, black arrowhead). Epithelial cells appeared to be dying (**c**, blue arrow). (**B**) Quantification of membranous structures via the point-counting method. LER, lamellarly/concentrically arranged endoplasmic reticula; Ly, lysosomes. The number of fields totaled 30 from each mouse. The results represent the mean, SEM. *** *p* < 0.001.

**Figure 4 ijms-20-01711-f004:**
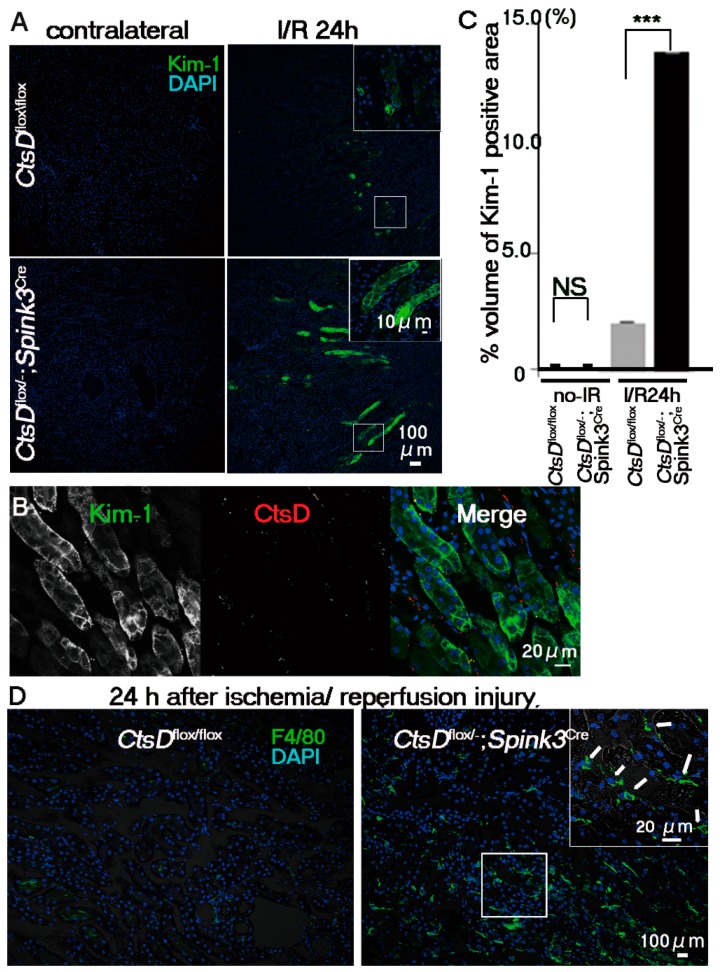
*CtsD*-deficiency caused severe damage in renal proximal tubular epithelial cells following ischemia/reperfusion injury. (**A**) Immunostaining of Kim-1, a marker of early acute injury of renal tubular cells (green) with DAPI (blue) in renal tissues obtained from *CtsD^flox/flox^* and *CtsD^flox/−^; Spink3^Cre^* mice at 24 h after ischemia/reperfusion injury (I/R 24 h). As control sample, the cortico-medullary region of the contralateral kidney was employed (contralateral). Scale bars indicate 100 μm (**B**) Representative image of Kim-1 (green) and cathepsin D (CtsD; red) immunostaining in *CtsD ^f/−^*; *Spink3^Cre^* mouse kidney, 24 h after ischemia/reperfusion injury. DAPI (blue) (**C**) Quantification of Kim-1-positive areas of the cortico-medullary region of *CtsD^flox/flox^* and *CtsD^flox/−^; Spink3^Cre^* mice kidneys. The volume density of Kim-1-positive renal proximal tubular epithelial cells in total renal tissues obtained from *CtsD^flox/−^; Spink3^Cre^* (black, *CtsD^flox/−^; Spink3^Cre^* I/R 24 h; 13.7 ± 7 × 10^−5^%) and *CtsD^flox/flox^* (*CtsD^flox/flox^* I/R24h; 2.12 ± 2 × 10^−5^%) mice at 24 h following ischemia/reperfusion injury. n = 3, *** *p* < 0.001. In kidneys without I/R injury, there was little difference between *CtsD^flox/flox^* (*CtsD^flox/flox^* no-I/R; 0.02 ± 5 × 10^−7^%) and *CtsD^flox/−^; Spink3^Cre^* mice (*CtsD^flox/−^; Spink3^Cre^* no-I/R; 0.04 ± 9 × 10^−7^%). n = 3 (**D**) Representative image of kidney sections immunostained with F4/80, a marker for macrophage (green) and DAPI (blue), at 24 h after renal ischemia reperfusion injury. Signals of F4/80 increased in *CtsD^flox/−^; Spink3^Cre^* kidney (white arrows). Scale bars indicate 100 μm and 20 μm in insert.

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
