# Peer review of "Lack of Cathepsin D in the Renal Proximal Tubular Cells Resulted in Increased Sensitivity against Renal Ischemia/Reperfusion Injury"

_ijms, 2019, doi:10.3390/ijms20071711_

Reviewer 1 Report

This manuscript reports a study carried out in a conditions knockout CtsD-flox/-; Spink3-Cre mouse aiming at investigating the role of chatepsin D in the renal proximal tubular cells. The results provided in the study indicate that this mouse present significant reduction in the level of cathepsin D in tubular epithelial cells of the cortico-medullary region of the kidney. This deficiency is associated witha marked increase in LC3, a marker of autophagosome/autolysosome signal. Furthermore, induction of renal ischemia/reperfusion injury resulted in the increase of the early injury marker Kim-1. The conclusion of the authors is that lack of cathepsin D in renal tubular epithelial cells led to an increase of sensitivity against ischemia-reperfusion injury.

Author Response

Thank you for your effort and great comments for our manuscript.

We wish to express our appreciation to you for positive comments on our manuscript.

Reviewer 2 Report

Comments to the authors:

The authors, using the a conditional mouse model CtsDflox/- Spink3Cre, demonstrate that cathepsin D exhibits a cytoprotective role in renal proximal tubular epithelial cells after ischemia/ reperfusion injury.

Knockout mice show an increase of LC3 exression levels and after renal ischemia/reperfusion injury Kim-1 is incresed in the ko mice kidney. The authors conclude that the absence of renal cathepsin D increases the sensitivity against ischemia/ reperfusion injury.

The paper is well written and scientifically sound, however, several points should be consider:

Major comments

1- the authors should add details about the conditional mouse model and the 

protocol used for the genetic Cts deletion.

2- the authors should also perform the western blotting experiments to quantify the increase of LC3 and Kim-1 expression in kidney of CtsDflox/- Spink3Cre compared to control mice. 

3- in Fig 4A the authors showed the Kim-1 expression in CtsDflox/- Spink3Cre. They also showed this result in fig 4B. I think that they should remove it.

4- the discussion is too short and should be implemented. In the results section there are sentences which can be moved to the discussion, for example page 3 lanes 80-82, page 4 lanes 95-99, page 5 lanes 133-135.

5- What is the difference between fig 1 and fig S3 B?

6- the authors should indicate the statistical analysis performed on urine and blood data reported in Fig S1.

Minor comments

1- Page 1 The Authors  shouldexplicit LC3 and Kim-1 acronyms

2- Page 1 lane 43-44 insert references

3- Page 6 lane In the Fig 4 C use the same nomenclature for CtsDflox/- Spink3Cre

4- the authors should indicatespecify the genotype of wild type mice used as controls in the experiments.

Author Response

We appreciate your comments and suggestions. Our responses to your comments are as follows:

Major comments

Comment 1- the authors should add details about the conditional mouse model and the protocol used for the genetic Cts deletion.

 Response: Following the reviewer’s comments, we added an explanation in the manuscript (page 5 lines 68 to 74).

Comment 2- the authors should also perform the western blotting experiments to quantify the increase of LC3 and Kim-1 expression in kidney of CtsDflox/- Spink3Cre compared to control mice. 

Response: During our investigation, we used whole kidney lysates of CtsDflox/- Spink3Cre and CtsDflox/flox mice for immunoblot analysis of LC3 and Kim-1. However, we could not detect a significant difference between the two, most likely because cells of cortico-medullary correspond to a small fraction of all cells in the kidney. We followed up our analysis by isolating renal proximal tubular cells from the mouse kidney by using FACS. However, we could not perform immunoblot analysis of LC3 and Kim-1, since only a small amount of renal cells was isolated. Therefore, to quantify the increase of LC3 and Kim-1 we relied on Figure 2A-B and Figure 4A-C as we described in our manuscript.

Comment 3- in Fig 4A the authors showed the Kim-1 expression in CtsDflox/- Spink3Cre. They also showed this result in fig 4B. I think that they should remove it.

 Response: In Fig 4A we show the increment of Kim-1 after ischemia reperfusion injury. In Fig 4B we show how Kim-1 is expressed in the cathepsin D deleted tubular cells. We consider that they are showing different results. However, we acknowledge that the figures may cause some confusion, therefore we added the following explanation in the main text (page 14 lines 179 to 180): “It indicates that cathepsin D is deficient in the same tubular cells as the kim-1 expressed cells”.

Comment 4- the discussion is too short and should be implemented. In the results section there are sentences which can be moved to the discussion, for example page 3 lanes 80-82, page 4 lanes 95-99, page 5 lanes 133-135.

 Response: We moved the sentences pointed out by the reviewer and improved the text for our discussion.  These parts are written in red colored font in the Discussion section. (page 18, lines 220 to222 , 226 to page 19, line230, and page19, line244 to page 20, line 252)

Comment 5- What is the difference between fig 1 and fig S3 B?

 Response: Fig S3 B shows higher magnification of fig1. We acknowledge that it causes confusion, therefore, we deleted fig S3 B.

Comment 6- the authors should indicate the statistical analysis performed on urine and blood data reported in Fig S1.

Response: Following the reviewer’s comment, we improved the text in those legends as follows: ’Data in graphs are expressed as the mean ± SEM. Statistical comparisons between the groups were performed using a student’s t-test, and statistical significance was set at p<0.05.’< strong="">. Additionally, we added p values in graphs for Fig S1 and FigS2.

Minor comments

Comment 1Page 1 The Authors should explicit LC3 and Kim-1 acronyms

 Response: In response to this comment we described our acronyms as’microtubule-associated protein light chain 3 (LC3; a marker for autophagosome/ autolysosome)‘ and ‘Kidney injury molecule 1 (Kim-1)’ (page 2, lines 30, 31, 33) We also added an Abbreviations section in the main text. (page 25)

Comment 2Page 1 lane 43-44 insert references

 Response: As the reviewer suggested, we inserted the references for these sentences.

Comment 3Page 6 lane In the Fig 4 C use the same nomenclature for CtsDflox/- Spink3Cre

 Response: Following the comment 3, we corrected Fig 4C.

Comment 4- the authors should indicate specify the genotype of wild type mice used as controls in the experiments.

 Response: We used CtsDflox/flox mouse for control. We described it in 4.1 Animal model (page 21 line 267) ‘‘CtsD flox/flox littermates that served as the control.’ But for clarity, we changed the description in the all figures form ‘Control’ to ‘CtsDflox/flox’.

Round  2

Reviewer 2 Report

The manuscript has been significantly improved and now warrants publication in IJMS.